# Dietary Differentiation Mitigates Interspecific Interference Competition Between Sympatric Pallas’s Cats (*Otocolobus manul*) and Red Foxes (*Vulpes vulpes*)

**DOI:** 10.3390/ani15091267

**Published:** 2025-04-29

**Authors:** Dong Wang, Quanbang Li, Jingyu Gao, Luyi Hou, Yanjun Zou, Xinming Lian

**Affiliations:** 1Key Laboratory of Adaptation and Evolution of Plateau Biota, Northwest Institute of Plateau Biology, Chinese Academy of Sciences, Xining 810008, China; wangdong@nwipb.cas.cn (D.W.); liquanbang@nwipb.cas.cn (Q.L.); gaojingyu24@mails.ucas.ac.cn (J.G.); houluyi@nwipb.cas.cn (L.H.); zouyanjun@nwipb.cas.cn (Y.Z.); 2School of Geographical Science, Qinghai Normal University, Xining 810016, China; 3University of Chinese Academy of Sciences, Beijing 100049, China; 4Qinghai Provincial Key Laboratory of Animal Ecological Genomics, Xining 810008, China

**Keywords:** coexistence, trophic niche, dietary composition, Pallas’s cat (*Otocolobus manul*), red fox (*Vulpes vulpes*), Sanjiangyuan National Park

## Abstract

The differential utilization of food resources plays a critical role in alleviating the intensity of competition among sympatric carnivorous species. In the Sanjiangyuan National Park (SNP), there are 13 species of small carnivores (body mass < 15 kg); however, the partitioning of trophic niches among these local carnivore species remains poorly understood. The sympatrically distributed small carnivores, the Pallas’s cat (*Otocolobus manul* Palls, 1776) and the red fox (*Vulpes vulpes* Linnaeus, 1758), exhibit significant spatial overlap within the SNP, yet their dietary niche differentiation remains inadequately understood. To bridge this knowledge gap, we employed DNA barcoding technology to analyze the dietary composition and differences between these two predator species. Our findings demonstrate that the differential use of food resources is essential for mitigating interspecific competition among local small carnivores and promoting regional coexistence.

## 1. Introduction

Small carnivores (body mass < 15 kg) act as key primary consumers within ecosystems, playing a crucial role in processes such as energy flow and substances cycling [1]. Through predation, competition, and trophic cascading effects, they significantly influence the structure and function of ecosystems [2,3]. In ecosystems where large carnivores are absent, small carnivores functionally assume the role of apex consumers, exerting critical regulatory control over ecosystem stability and functional persistence [3,4]. As opportunistic predators, they typically consume a diverse array of prey resources, leading to complex dietary compositions and broader trophic niches [5].

Niche partitioning is anticipated to emerge when co-occurring species engage in direct competition for identical resources; however, intraguild interactions among predator species may still occur even in the absence of resource limitation [6]. For carnivores coexisting within the same geographic region, species with comparable body sizes and significant dietary overlap typically experience the most intense competition [7], which may lead to interference competition and resource utilization competition [8,9]. To alleviate interspecific competition, sympatric carnivores often achieve coexistence by differentially utilizing prey resources, altering feeding proportions, adjusting activity ranges, and modifying the onset and cessation times of their activities or behavioral rhythms [10]. Among sympatric carnivores, trophic niche differentiation represents the predominant mechanism that mediates the reduction of interspecific competition across spatial, temporal, and trophic dimensions [11]. Despite dietary spectrum overlaps in sympatric carnivore assemblages, divergent prey selection strategies and resource utilization patterns promote resource partitioning, thereby mitigating competitive exclusion through trophic niche complementarity [8].

Dietary strategy emerges as an adaptive mechanism in response to habitat limitations imposed by species-specific nutritional demands, while the trophic niche operationally delineates the multidimensional exploitation of food resources [12]. Niche breadth quantifies an organism’s trophic position and its involvement in interspecific interaction networks within ecosystems, with its dimensionality being structured by predation pressures and competitive exclusion dynamics mediated through species interactions [13,14]. Small carnivores predominantly function as opportunistic predators, with demonstrating complex dietary profiles characterized by diverse prey utilization patterns and broadening the trophic niche to enhance ecological fitness representing a pivotal mechanism for alleviating interspecific competition when competition escalates or during seasons of prey resource scarcity [15,16].

Traditional approaches to analyzing animal dietary habits have relied on morphological identification of undigested remains (e.g., hairs, skeletal fragments) in fecal samples [17]. These remains are compared against reference specimens or taxonomic keys of known species to infer prey composition [18,19]. However, due to the limitations imposed by food type and the degree of digestion, compounded by the presence of morphologically similar and closely related species in fecal matter, traditional methods demand analysts with extensive expertise in biological taxonomy. Consequently, this leads to reduced efficiency and accuracy in species identification [20]. In recent years, the advent of DNA barcoding technology has addressed many limitations of conventional methods for determining animal diets, enabling rapid and precise identification and taxonomic classification of species [21]. DNA barcoding is a technique that employs high-throughput sequencing to acquire large-scale amplicon sequences of various barcode genes from mixed samples. Subsequently, bioinformatics tools are utilized to analyze these sequences, facilitating the identification of taxonomic units [22,23]. Currently, DNA barcoding has been extensively applied in carnivore diet studies, allowing for both qualitative and quantitative descriptions of carnivores’ trophic niches, as well as the assessment of interspecific dietary overlap [24,25]. This technology offers critical technical support for investigating the mechanisms underlying sympatric species coexistence.

The Pallas’s cat (*Otocolobus manul* Palls, 1776) and the red fox (*Vulpes vulpes* Linnaeus, 1758) are two small carnivore species with comparable body sizes, belonging to the Felidae and Canidae families, respectively. They exhibit extensive and sympatric distribution within the Sanjiangyuan National Park (SNP), Qinghai Province, China. Previous studies have shown that these two species in the SNP demonstrate relatively high spatial overlap indices while maintaining distinct diurnal activity patterns [26]. However, there is a notable paucity of research examining trophic niche differentiation between these two small carnivores. The classical resource allocation hypothesis posits that sympatric carnivore species with morphological similarity experience heightened interspecific competition, which drives niche differentiation through divergent prey selection, foraging strategies, or the differential utilization of shared prey resources to mitigate competitive pressures [27]. To investigate the trophic niche differentiation between Pallas’s cats and red foxes within the SNP, this study utilized DNA barcoding technology to analyze their dietary compositions and overlap indices.

## 2. Materials and Methods

### 2.1. Study Area

The study was conducted in the Sanjiangyuan National Park (SNP; 32°26′04″~36°16′49″ N, 89°24′06″~99°06′46″ E), situated in the northeastern Qinghai-Xizang Plateau in China (Figure 1). The SNP serves as the source of three major rivers—the Yangtze River, the Yellow River, and the Lancang River—and encompasses three distinct geographical regions according to the three rivers [28]. The topography of the region is characterized by its complexity, with the average elevation exceeding 4500 m. The mean annual temperature ranges from –5.6 °C to 7.8 °C, and the cold season lasts for approximately seven months [29]. The SNP hosts a variety of ecological systems, including glacial formations, snow-capped peaks, high-altitude lakes and wetlands, alpine grasslands, and meadows. These diverse ecosystems support a rich and unique assemblage of plateau wildlife, earning the region the title of “Alpine Biological Germplasm Resource Bank” [30]. The biodiversity within the SNP is remarkable, comprising 760 species of vascular plants and 125 species of wildlife [28,31].

### 2.2. Fecal Sample Collection

This study collected a total of 160 suspected carnivore fecal samples via route search method in the SNP and its surrounding regions during December 2023 (cold season) and July 2024 (warm season). To minimize the possibility of multiple fecal samples originating from the same individual, during fecal sample collection, we ensured that the sampling distance between samples with similar external morphologies was greater than 1 km, with a range up to 1039.28 km, and that the mean sampling distance was 155.41 ± 1.51 km (mean ± SEM), thereby reducing potential sampling bias. Upon encountering relatively fresh fecal samples, GPS coordinates and sampling time were meticulously documented. Researchers subsequently collected appropriate quantities of fecal material using disposable polyethylene (PE) gloves, transferring the samples into 50 mL sterile tubes labeled with collection dates and unique identification codes. Disposable gloves were replaced between each sampling event to prevent cross-contamination. The collected samples were immediately submerged in absolute ethanol on the day of collection. After 24 h of fixation, the ethanol was carefully decanted and replaced with silica gel desiccants for moisture absorption. All specimens were subsequently transported to the laboratory and cryopreserved at −20 °C until further analysis. Strict adherence to standardized protocols throughout the sampling process ensured sample integrity and minimized environmental DNA degradation.

### 2.3. Fecal DNA Extraction, Amplification and High-Throughput Sequencing

This study employed the QIAamp Fast DNA Stool Mini Kit (QIAGEM, Hilden, Germany) for extracting DNA from fecal samples, and the Qubit assay was used to quantify the DNA concentration.

The host species and dietary composition of fecal samples were identified using 12S V5-F/R primers [32], with the PCR reaction system referenced from prior studies [25]. In this study, PCR products were purified using the E.Z.N.A. Gel Extraction Kit (Omega, NJ, USA). Subsequently, paired-end sequencing (250 bp read length) was carried out on the Illumina Nova-Seq 6000 sequencing platform. Library preparation and sequencing were performed by Guangdong Magigene Biotechnology Co., Ltd. (Guangzhou, China). Primer details are provided in Table 1.

### 2.4. Data Processing and Analysis

The reads obtained from second-generation sequencing were subjected to initial quality control using the Fastp software (version 0.12.4). This process included adapter trimming and the filtering of low-quality reads. The filtered paired-end reads were subsequently assembled using the USEARCH software (version 11.0.667), with a default selection of 350,000 reads or all reads if the total number was below this threshold. The VSEARCH software (version 2.8.1) was employed to remove primer sequences by aligning them at both ends and discarding sequences with more than 1% sequencing errors. Additionally, the length of the assembled sequences was restricted to the specified range of the amplified product to exclude non-target-length sequences. All assembled sequences from the samples were then pooled, and duplicate sequences were removed using the VSEARCH software. Only sequences with an abundance greater than four were retained as representative sequences. The chimera-filtered representative sequences were further processed using the uchime3_denovo algorithm in USEARCH software (version 5.2.32) for de novo chimera removal. Subsequently, Operational Taxonomic Unit (OTU) clustering was performed using selected OTU delineation algorithms (UCLUST/UPARSE/UNOISE) to generate OTU representative sequences. Finally, the usearch_global algorithm in USEARCH was employed to align the merged assembled sequences against the OTU sequence set, retaining sequences with >97% coverage and >99% similarity to construct the final OTU table.

The obtained OTU sequences were aligned against the reference sequences of the target gene metabarcoding database using BLAST software (V 2.2.31). Sequences with an alignment length of less than 90% and similarity below 80% were excluded. Taxonomic classification was hierarchically assigned at five levels (species, genus, family, order, and class) based on default similarity thresholds of 98%, 95%, 90%, and 85%, respectively. The annotation results were appended to the final column of the OTU table. Subsequently, the final taxonomic assignments were determined by evaluating candidate species for each OTU. Host-derived OTU sequences were removed, and target taxonomic groups were selected. Additionally, prey species identification was validated against the SNP species distribution inventory to confirm the regional presence of detected species.

(1)This study utilized relative read abundance (*RRA*%) to quantitatively describe the proportion of sequence abundance for each prey taxonomic group in fecal samples following weighted averaging. The calculation formula is as follows:$${RRA}_{i}=\frac{1}{S}\sum_{k=1}^{s}\frac{n_{i,k}}{\sum_{i=1}^{T}n_{i,k}}\times 100\%$$
In this formula, the parameter *S* denotes the total number of fecal samples collected for the target species, *T* represents the total number of prey species identified, and *n_i,k_* signifies the number of sequences attributed to prey species *i* within fecal sample *k*. The cumulative relative read abundance (*RRA*) for each prey species across all fecal samples of a given target species equals 100%.(2)To further investigate the dietary differences between the two small carnivores, this study employed the Levins index (*B*) and the Pianka overlap index (*O_jk_*). These indices were applied to evaluate the dietary niche breadth of the Pallas’s cat and the red fox [33], as well as the extent of dietary overlap between them [34,35], based on the relative abundance of food composition at the species level.$$B=\frac{1}{\sum_{i=1}^{n}P_{j}^{2}}$$$$O_{jk}=\frac{\sum P_{ij}\cdot P_{ik}}{\sqrt{\sum {P_{ij}}^{2}\cdot \sum {P_{ik}}^{2}}}$$
In the above formula, parameter *B* denotes niche breadth, while *P_j_* signifies the proportion of food item *j*. The value of *B* ranges from 0 to infinity. *O_jk_* represents the Pianka overlap index, where *P_ij_* and *P_ik_* indicate the relative abundance of food item *i* in the diets of species *j* and *k,* respectively. The range of *O_jk_* values extends from 0 to 1. According to the evaluation criteria, an *O_jk_* value of 0 implies no dietary overlap, whereas an *O_jk_* value of 1 suggests complete dietary overlap. An *O_jk_* value exceeding 0.3 is deemed to signify significant overlap, and an *O_jk_* value surpassing 0.6 is considered to reflect pronounced overlap.

## 3. Results

Through DNA extraction, sequencing, and sequence alignment of the collected fecal samples, 26 samples were confirmed to belong to Pallas’s cats, while 13 were identified as originating from red foxes. The average altitude of collection sites for the 26 Pallas’s cat fecal samples was 4799 m (ranging from 4690 m to 4984 m), whereas the average altitude for the 13 red fox fecal samples was 4768 m (ranging from 4456 m to 5051 m).

### 3.1. Dietary Composition of Pallas’s Cats

A total of 10 prey species were identified in the fecal samples of Pallas’s cats, comprising four mammal species (*RRA* = 82.83%), four bird species (*RRA* = 1.39%), and two livestock species (*RRA* = 1.41%) (Figure 2 and Figure 3). Additionally, a significant proportion of the diet consisted of unidentified species within the family Cricetidae (*RRA* = 14.39%). Within the identified prey composition of Pallas’s cats, the top three mammalian species ranked by relative abundance, in descending order, were the plateau pika (*Ochotona curzoniae* Hodgson, 1858; *RRA* = 69.34%), Himalayan marmot (*Marmota himalayana* Hodgson, 1841; *RRA* = 12.10%), and Blyth’s mountain vole (*Neodon leucurus* Blyth, 1863; *RRA* = 1.17%). Collectively, these species accounted for 82.61% of the total relative abundance (Figure 2). Avian species constituted a minimal proportion within the dietary composition of Pallas’s cats, with all individual avian prey species exhibiting relative abundances below 1.00%. Statistical analysis revealed that avian prey collectively accounted for only 1.39% of the total relative abundance in the cats’ dietary profile (Figure 2).

### 3.2. Dietary Composition of Red Foxes

A total of 18 prey species were identified in the fecal samples of red foxes, comprising nine mammal species (*RRA* = 63.55%), six bird species (*RRA* = 10.43%), and three livestock species (*RRA* = 23.58%) (Figure 4 and Figure 5). Additionally, the relative abundance of unidentified species within the Cricetidae family was determined to be 2.44%. Within the dietary composition of the red fox, the top three mammalian prey species by relative abundance, in descending order, were the plateau pika (*RRA* = 40.63%), domestic yak (*Bos grunniens* Linnaeus, 1978; RRA = 23.00%), and woolly hare (*Lepus oiostolus* Hodgson, 1848; *RRA* = 9.64%). These three prey species collectively accounted for 73.27% of the total dietary relative abundance, while the remaining mammalian species contributed only 4.03% (Figure 4). Avian species collectively constituted 10.43% of the dietary composition in red foxes, with the three species exhibiting the highest relative abundances being the common buzzard (*Buteo japonicus* Linnaeus, 1758; *RRA* = 7.53%), horned lark (*Eremophila alpestris* Linnaeus, 1758; *RRA* = 2.23%), and carrion crow (*Corvus corone* Linnaeus, 1758; *RRA* = 0.27%), listed in descending order of prevalence (Figure 4).

### 3.3. Dietary Breadth and Overlap Between Pallas’s Cats and Red Foxes

Statistical analysis revealed that mammals constituted the predominant prey item for Pallas’s cats, accounting for 97.20% of their diet (Figure 6). In contrast, the diet of red foxes comprised a significant proportion of livestock and birds in addition to mammals (Figure 6). The calculations of dietary niche breadth and overlap indicated that the red fox (*B* = 267.89) exhibited a broader dietary niche compared to the Pallas’s cat (*B* = 162.94). Additionally, despite a high dietary niche overlap coefficient between Pallas’s cats and red foxes (*O_jk_* = 0.81), significant differences were observed in their feeding proportions of different species (*p* = 0.0123). This indicates that Pallas’s cats and red foxes primarily mitigate interspecific competition by preying on different species and proportions.

## 4. Discussion

The dietary composition of animals serves as a critical reflection of their ecological relationships with habitats and, to a certain extent, acts as an indicator of local species diversity [36]. Investigating feeding ecology is essential for understanding organism–environment interactions and predator–prey dynamics, while also forming a fundamental basis for addressing contemporary scientific priorities. These include the development of habitat utilization models, the investigation of disease transmission mechanisms, the assessment of species survival pressures, and biodiversity evaluation [37,38]. DNA barcoding technology has been extensively utilized in the dietary analysis of carnivores, encompassing species such as brown bears (*Ursus arctos* Linnaeus, 1758), lynxes (*Lynx lynx* Linnaeus, 1758), and leopard cats (*Prionailurus bengalensis* Kerr, 1792), facilitating precise elucidation of regional food web construction and interspecific relationships [39,40].

Constrained by limited environmental resources, competitors coexisting in the same habitat have gradually evolved distinct foraging strategies over prolonged evolutionary timescales to fulfill their survival requirements [41]. Trophic resource partitioning, as a critical dimension of interspecific competition, plays a key role in elucidating the coexistence mechanisms of sympatric species within shared ecological niches [15]. This study employed DNA barcoding technology to quantify the dietary composition and interspecific overlap between two sympatric species, the Pallas’s cat and the red fox. It elucidated the mechanisms by which these two species achieve sympatric coexistence through the lens of nutritional niche differentiation. Moreover, the study highlighted the efficiency and convenience of utilizing fecal DNA for rapid species surveys. Investigating the mechanisms underlying regional coexistence from spatial and temporal niche dimensions revealed that Pallas’s cats and red foxes exhibit relatively high spatial overlap in habitat utilization, but mitigate interspecific competition by reducing temporal overlap in activity patterns [28]. Our study also addressed the limitations of previous research that focused solely on the spatial and temporal niche perspectives of local coexistence mechanisms for the Pallas’s cat and the red fox, without considering dietary niche differentiation. In the context of the SNP, despite a high degree of dietary overlap between the Pallas’s cat and the red fox (*O_ij_* = 0.81), these two species achieve sympatric coexistence primarily through differential feeding proportions on shared prey species, thereby alleviating interspecific competition. Additionally, the red fox (*B* = 267.89) exhibited a broader trophic niche breadth compared to Pallas’s cat (*B* = 162.94), with significantly greater prey diversity in its diet. Collectively, the dietary differences between the two small carnivore species also provide validation for the resource allocation hypothesis. These ecological differences enhance trophic niche divergence between the two carnivores, mitigating interspecific interference and competition, and thus promoting regional coexistence via resource partitioning mechanisms.

Research indicated that small carnivores predominantly function as opportunistic predators, with their primary prey consisting of small mammals, birds, amphibians, reptiles, and invertebrates. These species exhibit omnivorous tendencies to varying degrees, thereby occupying relatively broad trophic niches [42,43]. Moreover, due to their greater tolerance for human disturbances compared to large carnivores, small carnivores may also consume food resources associated with human activities [44]. Previous studies have demonstrated that more than 50% of the Pallas’s cat’s dietary composition consists of small mammals [24]. Additionally, Pallas’s cats inhabit rocky crevices or abandoned marmot burrows year-round [26]. Our study revealed that Pallas’s cats predominantly prey on plateau pikas (*RRA* = 69.34%), with Himalayan marmots constituting a secondary yet significant component of their diet (*RRA* = 12.10%). These findings corroborate previous studies that have identified plateau pikas and small rodents as the primary prey for Pallas’s cats, while also emphasizing the critical role of predation pressure from small carnivores like Pallas’s cats in maintaining population stability within pika colonies across their distribution areas [24,45,46]. This trophic regulation mechanism underscores the ecological significance of meso-predators in maintaining grassland ecosystem stability through top-down control of herbivore populations. Dietary analysis revealed that red foxes predominantly consumed plateau pikas (*RRA* = 40.63%) and domestic yaks (*RRA* = 23%), which together accounted for the largest proportion of their diet. Additionally, traces of large ungulate prey, such as kiang, Tibetan antelope, and bharal, were detected in their dietary composition. This phenomenon is consistent with findings from other studies [24,46]. It implies that by consuming a certain proportion of large ungulate prey, the red fox can effectively alleviate competition with other sympatric small carnivores. However, considering that red foxes typically hunt individually, it is unlikely that they actively pursue these large prey species. It is hypothesized that red foxes frequently follow the activities of sympatric large carnivores to scavenge on carcasses left over from their predation [47,48]. Kleptoparasitism also serves as a critical survival strategy. For instance, smaller coyotes (*Canis latrans* Say, 1823) tracked the movements of gray wolves (*Canis lupus* Linnaeus, 1758) to exploit foraging opportunities by feeding on carcasses killed by the latter. This low-energy foraging strategy mitigated the predation risks posed by gray wolves to coyotes [49]. Additionally, studies have highlighted that species such as red foxes and Tibetan foxes follow the activities of brown bears to capture individuals that escape when brown bears excavate burrows to prey on plateau pikas [50]. Consequently, kleptoparasitism effectively elucidates the presence of large ungulate prey in the diet of small carnivores like red foxes and explains the coexistence of small and large carnivores within the same ecological area. In addition to acquiring carcasses of large prey through kleptoparasitism, we hypothesize that red foxes may also encounter naturally deceased large ungulates during their foraging activities in natural habitats. Together, these two ecological mechanisms help explain the presence of multiple large prey species in the dietary composition of red foxes.

In addition to domestic yak and sheep, the fecal samples of two small carnivore species also revealed traces of domestic pig, a non-native husbandry species. This finding could potentially be explained by the consumption of anthropogenic food waste containing remnants of domestic pigs, which were discarded by pastoralists during grazing activities and subsequently scavenged by these carnivores. Moreover, both Pallas’s cat (*RRA* = 14.39%) and red fox (*RRA* = 2.44%) exhibited unidentified Cricetidae species in their dietary composition. A literature review indicated that the local Cricetidae fauna comprised Tibetan dwarf hamster (*Urocricetus kamensis* Satunin, 1903), Stoliczka’s mountain vole (*Alticola stoliczkanus* Blanford, 1875), smoeky vole (*Neodon fuscus* Liu et al., 2012), Irene’s mountain vole (*Neodon irene* Thomas, 1911), and root vole (*Alexandromys oeconomus* Pallas, 1776) [51]. The unidentified Cricetidae sequences likely originate from these documented species within the study area. However, due to the limited number of fecal samples collected from Pallas’s cats and red foxes during the cold season, which lack sufficient statistical power, this study was unable to differentiate the seasonal impact on the dietary composition of these two small carnivores or examine variations in their feeding habits between the cold and warm seasons. Consequently, further collection of fecal samples across multiple seasons is necessary to evaluate the trophic niche differentiation between Pallas’s cats and red foxes with greater resolution.

## 5. Conclusions

The trophic niche, as a critical dimension of ecological niches, plays a pivotal role in shaping species-specific functional roles within ecosystem networks and their associated food web architectures [52]. A comparative analysis of dietary divergence among sympatric species elucidates patterns of trophic niche differentiation, which serves as a critical metric for quantifying potential interspecific competition intensity via niche overlap indices [53]. Our study demonstrated that Pallas’s cats and red foxes predominantly prey on plateau pikas and other small rodents within the SNP. Despite a high degree of trophic niche overlap between these two small carnivores, interspecific competition is mitigated through differential feeding proportions on shared prey. Additionally, red foxes exhibit greater prey diversity compared to Pallas’s cats, with a broader trophic niche breadth. These factors collectively reduce interspecific competition and interference, thereby promoting sympatric coexistence between Pallas’s cats and red foxes. The SNP exhibits high population density of plateau pikas (averaging 300–412 individuals/ha), which serve as primary prey for Pallas’s cats and red foxes [54,55], alongside abundant Himalayan marmots. This substantial prey availability partially mitigates interspecific competition between these two small carnivores. This research enhances our understanding of the potential intensity of interspecific competition and the mechanisms underpinning regional coexistence among sympatric small carnivore species within the SNP, thereby offering a scientific foundation for informed conservation and management strategies regarding these species.

## Figures and Tables

**Figure 1 animals-15-01267-f001:**
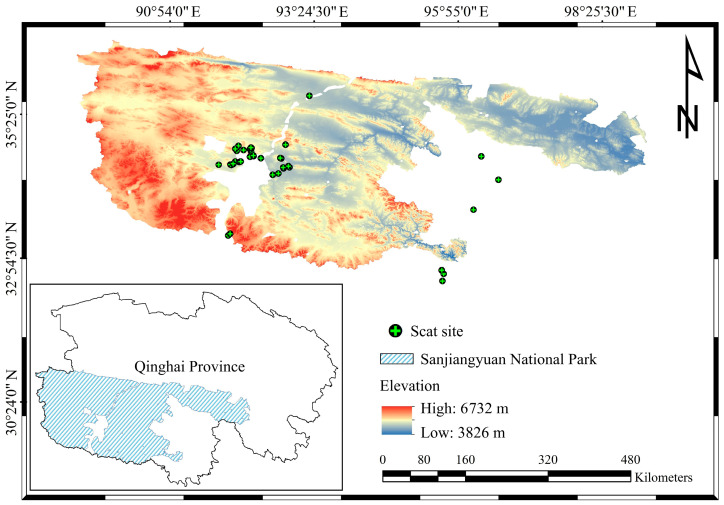
Locations of carnivore feces collected in Sanjiangyuan National Park.

**Figure 2 animals-15-01267-f002:**
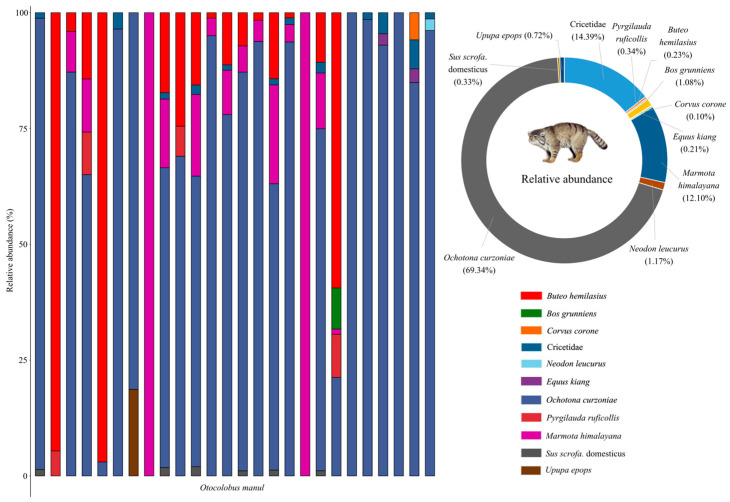
The dietary composition and corresponding relative abundance of prey for Pallas’s cats in the Sanjiangyuan National Park.

**Figure 3 animals-15-01267-f003:**
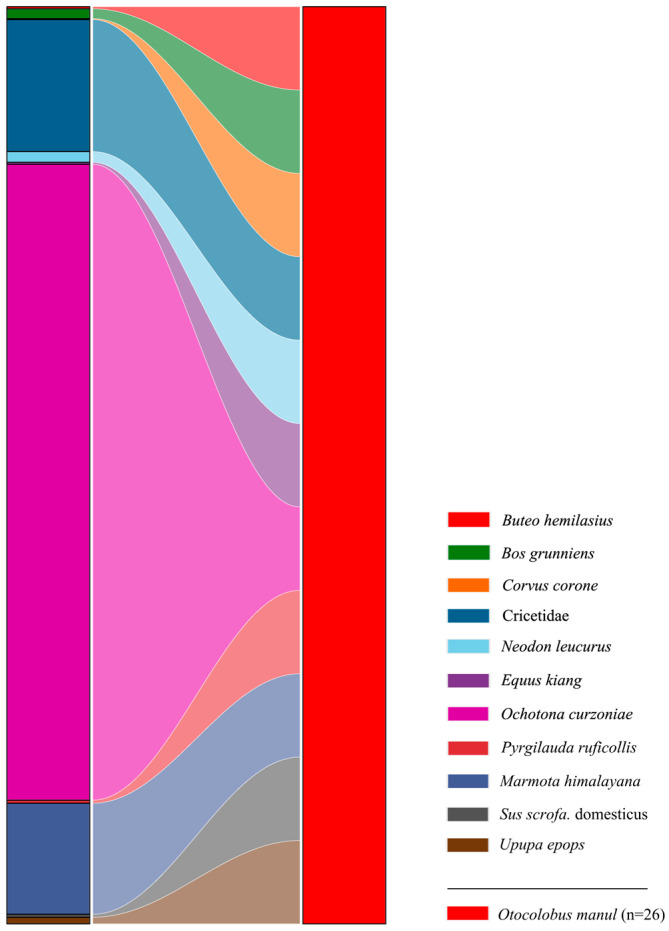
A Sankey diagram at the species level depicting the dietary composition of Pallas’s cats in the Sanjiangyuan National Park. On the left, color blocks represent the relative abundance of prey items classified at the species level, while on the right, color blocks denote the number of fecal samples collected from Pallas’s cats, with block sizes proportional to sample quantities. Numeric values in parentheses specify the quantities of fecal samples from Pallas’s cats.

**Figure 4 animals-15-01267-f004:**
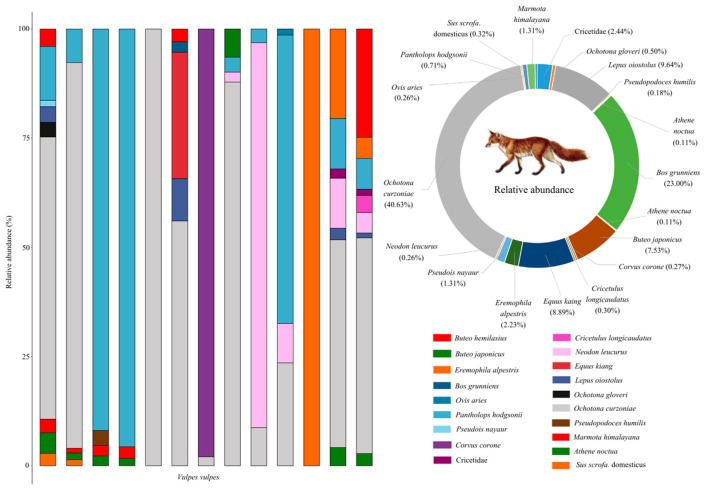
The dietary composition and corresponding relative abundance of prey for red foxes in the Sanjiangyuan National Park.

**Figure 5 animals-15-01267-f005:**
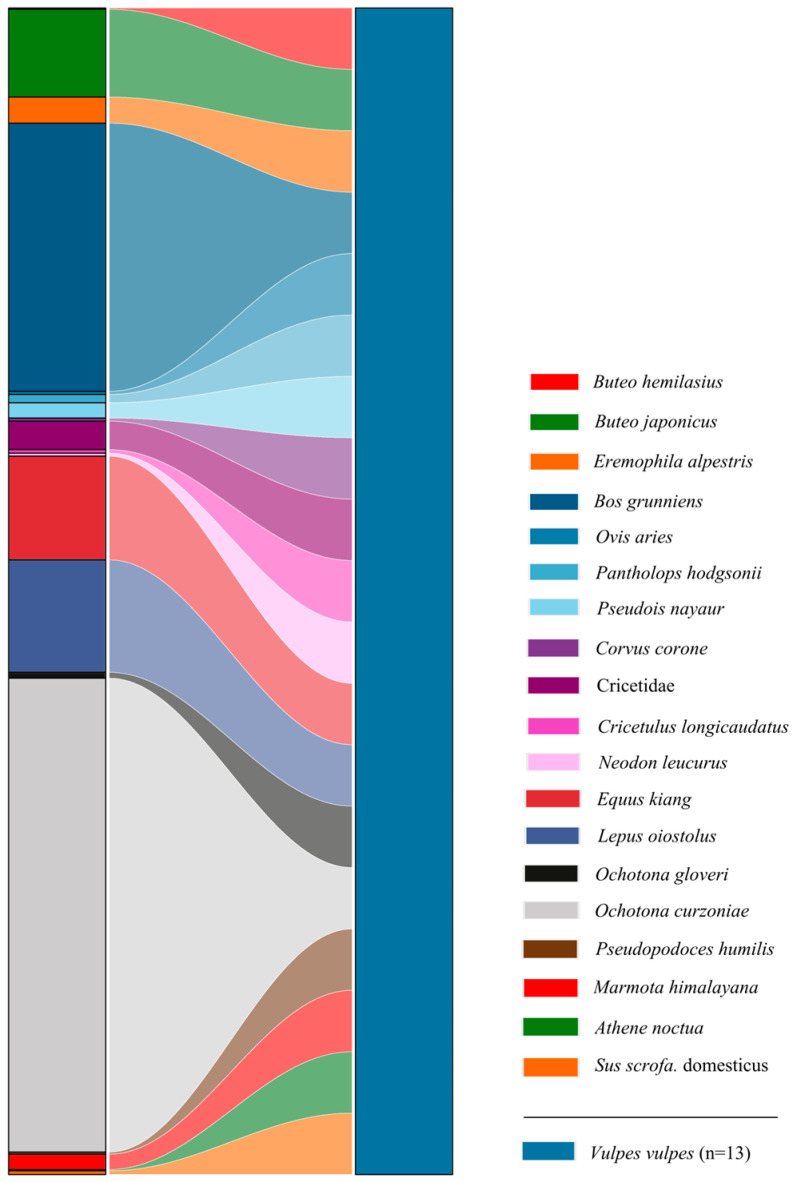
A Sankey diagram at the species level depicting the dietary composition of red foxes in the Sanjiangyuan National Park.

**Figure 6 animals-15-01267-f006:**
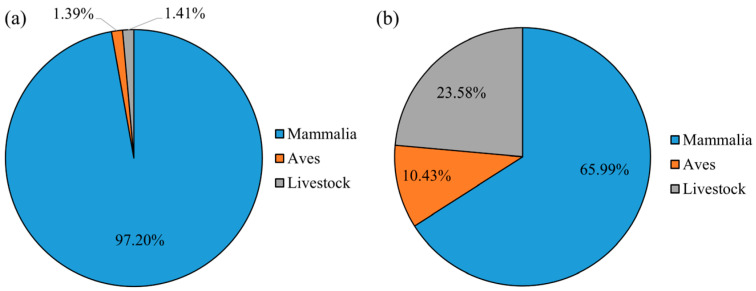
The proportions of mammals, birds, and livestock in the dietary composition of Pallas’s cat (**a**) and red fox (**b**) in Sanjiangyuan National Park.

**Table 1 animals-15-01267-t001:** Amplification primer information sheet.

Primer Pair	Sequences	Target Region
F_12SV5	5′-TAGAACAGGCTCCTCTAG-3′	12S
R_12SV5	5′-TTAGATACCCCACTATGC-3′	

## Data Availability

Data will be made available on request.

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
