# Peer review of "Dietary Differentiation Mitigates Interspecific Interference Competition Between Sympatric Pallas’s Cats (Otocolobus manul) and Red Foxes (Vulpes vulpes)"

_animals, 2025, doi:10.3390/ani15091267_

Round 1
Reviewer 1 Report
Comments and Suggestions for Authors
This study investigates the coexistence mechanisms of two sympatric small carnivores, the Pallas's cat (Otocolobus manul) and the red fox (Vulpes vulpes), in the Sanjiangyuan National Park of Qinghai Province, China. The research is conducted from a trophic niche perspective and provide compelling evidence regarding their interactions. The article is characterized by its rigor and features comprehensive discussions; furthermore, the findings offer valuable insights for further studies on competition and coexistence among sympatric species in this region. However, there are several minor issues in the article that require verification and revision by the authors.
Minor Revisions Suggested:
L27-L29: The abstract should include the number of fecal samples collected from both Pallas’s cats and red foxes.
L65, L80-L82: Please add appropriate references.
L103: The Latin scientific name Ochotona curzoniae should be italicized; additionally, at L219 and L233, this scientific name should be removed as it has already been mentioned in the Introduction section. It is advisable for the authors to conduct a thorough review of the manuscript to correct similar formatting errors throughout.
L152: Figure 2 should be replaced with Figure 1.
L251-255, L261-L265: The explanations provided for Figure 3 and Figure 5 are identical; it is recommended that only the captain for Figure 3 be retained.
L308-L321: The author needs to expand upon potential causes for dietary composition differences between Pallas’s cats and red foxes by integrating findings from previous studies with perspectives presented in this research.
L321-L330: While large ungulate prey within the dietary composition of red foxes appears primarily acquired through kleptoparasitism, I suggest acknowledging that a minor portion may also derive from scavenging naturally deceased ungulate carcasses found in nature. This potential source should be explicitly addressed by the authors in their discussion.
Additional Suggestions:
Discussion: The author should effectively integrate explanations into the discussion section regarding the factors contributing to the mitigation of interspecific competition between the two small carnivores, particularly from the perspectives of spatial and temporal niche differentiation.
In summary, I believe that this manuscript will achieve publishable standards upon the implementation of the aforementioned revisions.
Comments on the Quality of English LanguageThe English could be improved to more clearly express the research.
Author Response
Comments 1: L27-L29: The abstract should include the number of fecal samples collected from both Pallas’s cats and red foxes.
Response 1: Thank you for raising this point. We concur with your observation. As a result, we have incorporated the number of fecal samples collected from both Pallas’s cats and red foxes into the abstract. For further details, please refer to line 31 of the revised manuscript.
Comments 2: L65, L80-L82: Please add appropriate references.
Response 2: We sincerely appreciate your meticulous efforts. The corresponding references have been added to the relevant sections of the manuscript. For further details, please refer to lines 69 and 83 in the revised manuscript.
Comments 3: L103: The Latin scientific name Ochotona curzoniae should be italicized; additionally, at L219 and L233, this scientific name should be removed as it has already been mentioned in the Introduction section. It is advisable for the authors to conduct a thorough review of the manuscript to correct similar formatting errors throughout.
Response 3: We sincerely appreciate your meticulous attention to detail. We apologize for the aforementioned errors. To address this, we have conducted a thorough review of the Latin names of all species mentioned in the manuscript to ensure correct italicization. Additionally, we have removed redundant repetitions of Latin names for the same species throughout the document.
Comments 4: L152: Figure 2 should be replaced with Figure 1.
Response 4: We sincerely appreciate your meticulous attention to detail. We apologize for the aforementioned error. We have revised Figure 2 to Figure 1. For further details, please refer to line 160 in the updated manuscript.
Comments 5: L251-255, L261-L265: The explanations provided for Figure 3 and Figure 5 are identical; it is recommended that only the captain for Figure 3 be retained.
Response 5: We extend our gratitude for your valuable feedback. In accordance with your suggestions, we have eliminated the duplicate explanatory text for Figures 5 and 3, retaining solely the caption for Figure 3. For further details, please refer to lines 264-265 in the revised manuscript.
Comments 6: L308-L321: The author needs to expand upon potential causes for dietary composition differences between Pallas’s cats and red foxes by integrating findings from previous studies with perspectives presented in this research.
Response 6: We sincerely appreciate your meticulous work and valuable feedback. In accordance with your suggestions, we have refined the wording of this section to compare previous studies with our findings, thereby ensuring logical clarity. For further details, please refer to lines 322-339 in the revised manuscript.
Comments 7: L321-L330: While large ungulate prey within the dietary composition of red foxes appears primarily acquired through kleptoparasitism, I suggest acknowledging that a minor portion may also derive from scavenging naturally deceased ungulate carcasses found in nature. This potential source should be explicitly addressed by the authors in their discussion.
Response 7: We sincerely appreciate your valuable suggestion. In the Discussion section, we have explicitly stated that, in addition to being obtained through kleptoparasitism, the presence of large ungulate prey in the dietary composition of red foxes may also be attributed to carcasses of large animals that died naturally in the wild. For further details, please refer to lines 352-356 in the revised manuscript.
Comments 8: Additional Suggestions: The author should effectively integrate explanations into the discussion section regarding the factors contributing to the mitigation of interspecific competition between the two small carnivores, particularly from the perspectives of spatial and temporal niche differentiation.
Response 8: Thank you for your meticulous work and invaluable comments. We have supplemented the discussion with an exploration of the reasons for the regional coexistence of Pallas's cats and red foxes within the Sanjiangyuan National Park from the perspective of spatial and temporal niches. Additionally, we have explained that this study, which reveals the reasons for the regional coexistence of these two small carnivore species from the dimension of trophic niches, can address the limitations of previous research. Please refer to lines 300-304 for further details.
Response to Comments on the Quality of English Language
Point 1: The English could be improved to more clearly express the research.
Response 1: Thank you for your feedback regarding the quality of English in our manuscript. We appreciate your understanding and will strive to improve the clarity and precision of our language in the revised manuscript. All modifications throughout the manuscript are highlighted in red font.

Reviewer 2 Report
Comments and Suggestions for Authors
Overall, this manuscript is well organized and written, although the results are not at all surprising or new. It cites a Fig. 1, but none is presented...perhaps that is because it is the same as Fig. 2?? If so, this needs to be corrected. I'm not sure what Table 1 contributes...is it even needed? It seems there were only a small number of cat fecal samples (26), and a small number of foxes (13). This a small number upon which to determine food habits/diet. And could some of the fecal samples have been from the same individual, making the sample size even smaller? The last sentence in the conclusion is strange...saying the study can help with carnivore conservation without giving any examples of how that could occur.
Author Response
General comments: Overall, this manuscript is well organized and written, although the results are not at all surprising or new. It cites a Fig. 1, but none is presented...perhaps that is because it is the same as Fig. 2?? If so, this needs to be corrected. I'm not sure what Table 1 contributes...is it even needed? It seems there were only a small number of cat fecal samples (26), and a small number of foxes (13). This a small number upon which to determine food habits/diet. And could some of the fecal samples have been from the same individual, making the sample size even smaller? The last sentence in the conclusion is strange...saying the study can help with carnivore conservation without giving any examples of how that could occur.
Response: We sincerely appreciate your comprehensive and insightful feedback on our manuscript, which highlights your scientific rigor and has greatly contributed to the improvement of our work. Once again, we extend our gratitude for your valuable comments and the opportunity they have afforded us to refine our manuscript. Below are our explanations and modifications in response to the questions and concerns you raised.
Detailed comments:
Comments 1: It cites a Fig. 1, but none is presented...perhaps that is because it is the same as Fig. 2?? If so, this needs to be corrected.
Response 1: We extend our sincerest gratitude for your meticulous efforts. We sincerely apologize for the aforementioned error. We have corrected the reference from Figure 2 to Figure 1. For further details, please refer to the updated version of the manuscript at line 160.
Comments 2: I'm not sure what Table 1 contributes...is it even needed?
Response 2: We appreciate your valuable suggestion. In this study, the primers 12S V5-F/R were employed for identifying the host species of fecal samples and analyzing dietary composition. To provide comprehensive information, detailed primer amplification data have been included in Table 1, which can serve as a reference for similar studies. Consequently, Table 1 has been retained in the manuscript.
Comments 3: It seems there were only a small number of cat fecal samples (26), and a small number of foxes (13). This a small number upon which to determine food habits/diet. And could some of the fecal samples have been from the same individual, making the sample size even smaller?
Response 3: We sincerely appreciate your meticulous work and valuable feedback. As you pointed out, the relatively limited number of fecal samples from Pallas's cats and red foxes in this study might introduce potential biases in dietary analysis. Given the extreme challenges associated with collecting relatively fresh carnivore fecal samples in the field, we took every possible measure to ensure that the linear distance between adjacent fecal samples exceeded 1 km during field collection, thereby minimizing the likelihood that samples originated from the same individual. In future research, we will place significant emphasis on addressing the issue of sample size and strive to collect a sufficient number of fecal samples. Your feedback and suggestions are invaluable for guiding our future dietary studies.
Comments 4: The last sentence in the conclusion is strange...saying the study can help with carnivore conservation without giving any examples of how that could occur.
Response 4: We extend our sincerest gratitude for your meticulous work. In accordance with your revision suggestions, we have rephrased the final sentence of the conclusion to improve its academic rigor and readability. The revised sentence now reads as follows: This research enhances our understanding of the potential intensity of interspecific competition and the mechanisms underpinning regional coexistence among sympatric small carnivore species within the SNP, thereby offering a scientific foundation for informed conservation and management decisions regarding these species. For further details, please refer to lines 391-395.
Response to Comments on the Quality of English Language
Point 1: The English is fine and does not require any improvement.
Response 1: We sincerely appreciate your recognition of our entire manuscript. In response, we have conducted a comprehensive review of the full text and revised the unclear and erroneous sections to enhance the academic rigor and readability of the article. All modifications throughout the manuscript are highlighted in red font for easy identification.

Reviewer 3 Report
Comments and Suggestions for Authors
Dear Authors,
The manuscript touches upon the topic of overlapping trophic niches of the red fox and the Pallas's cat. The authors selected two species from different families, representatives of which often have a negative attitude towards each other. The choice of objects is insufficiently substantiated. This needs to be improved. The manuscript cannot be published in this form. The hypothesis is not formulated. It is not disclosed how the search for feces was conducted. It turned out that three quarters of the feces belong to completely different animal species. The size of the feces used for selection is not given. Accordingly, the entire sample of feces not related to the topic of the study should not be indicated. No attention is paid to the population density of predators in the study area. This also plays a role in determining the ecologic niche and, in particular, trophic relationships. Trends in the dynamics of the number of main food items are not given. There are no forecasts for changes in the predator-prey relationship. The text of the manuscript is mixed up in different chapters and should be moved to the appropriate chapters. In many methodological aspects, the authors omit important information. It should be added. The comparative part in the discussion needs to be expanded and additional literature sources on other canine and feline species should be cited. The focus should be on writing the conclusion of the manuscript based on the hypothesis. The authors have obtained interesting results, presented them, and they should be disclosed in the conclusions of the manuscript. After all the comments have been addressed, the manuscript can be reviewed again.

Author Response
General comments: The manuscript touches upon the topic of overlapping trophic niches of the red fox and the Pallas's cat. The authors selected two species from different families, representatives of which often have a negative attitude towards each other. The choice of objects is insufficiently substantiated. This needs to be improved. The manuscript cannot be published in this form. The hypothesis is not formulated. It is not disclosed how the search for feces was conducted. It turned out that three quarters of the feces belong to completely different animal species. The size of the feces used for selection is not given. Accordingly, the entire sample of feces not related to the topic of the study should not be indicated. No attention is paid to the population density of predators in the study area. This also plays a role in determining the ecologic niche and, in particular, trophic relationships. Trends in the dynamics of the number of main food items are not given. There are no forecasts for changes in the predator-prey relationship. The text of the manuscript is mixed up in different chapters and should be moved to the appropriate chapters. In many methodological aspects, the authors omit important information. It should be added. The comparative part in the discussion needs to be expanded and additional literature sources on other canine and feline species should be cited. The focus should be on writing the conclusion of the manuscript based on the hypothesis. The authors have obtained interesting results, presented them, and they should be disclosed in the conclusions of the manuscript. After all the comments have been addressed, the manuscript can be reviewed again.
Response: We sincerely appreciate your meticulous and insightful feedback on our manuscript, which underscores your scientific rigor and has greatly contributed to the improvement of our work. After carefully examining your suggestions, we have implemented substantial revisions, particularly in the methodology, discussion, and conclusion sections, to address the issues raised. We believe these modifications effectively address your concerns and enhance the comprehensiveness of our study. Once again, we extend our gratitude for your invaluable feedback and the opportunity it afforded us to refine our manuscript.
Detailed comments:
Comments 1: L17: how many?
Response 1: We sincerely appreciate your valuable feedback. After consulting relevant literature, we have ascertained that a total of 13 small carnivore species (with body mass < 15 kg) are distributed within the Sanjiangyuan National Park. For further details, please refer to lines 17-18 in the updated version of the manuscript.
Comments 2: Please indicate why you chose these two species of animals.
Response 2: We sincerely appreciate your insightful suggestion. In accordance with your recommendation, we have incorporated the rationale for species selection into the simplified summary. For further details, please refer to lines 19-23.
Comments 3: L30: This is not well connected to this sentence. I suggest that we state this in a separate sentence.
Response 3: We sincerely appreciate your meticulous work and invaluable comments. In response to your feedback, we have restructured this sentence into two statements to improve logical clarity. For further details, please refer to lines 29-32.
Comments 4: L45: substances?
Response 4: We extend our sincerest gratitude for your meticulous work. We sincerely apologize for this oversight. The error has been rectified by replacing ‘material’ with ‘substances’.
Comments 5: L59: and the beginning, the end of activity.
Response 5: We sincerely appreciate your insightful suggestion. In accordance with your recommendation, we have rephrased the relevant expression. For further details, please refer to lines 62-63.
Comments 6: L66-67: This should be rewritten.
Response 6: We sincerely appreciate your invaluable comments. In response to your feedback, we have restructured this sentence to improve its logical clarity. For further details, please refer to lines 70-71.
Comments 7: This is in Discussion.
Response 7: We sincerely appreciate your meticulous work and invaluable comments. In accordance with your suggestion, we have integrated the content from these sentences, which exhibited limited relevance to the introduction section, into the discussion section. For further details, please refer to lines 75-80.
Comments 8: L98: Please, at the first mention of a species in the text, write its full name (author, year).
Response 8: We sincerely appreciate your meticulous work. In accordance with your suggestion, we have revised the writing format for the first occurrences of all species throughout the entire text.
Comments 9: L98: It should be added that these are two species from different families (Canidae and Felidae), whose representatives often have a negative attitude towards each other.
Response 9: We sincerely appreciate your insightful suggestion. In accordance with your recommendation, we have added a clarification that Pallas's cat and the red fox belong to the families Felidae and Canidae, respectively. For further details, please refer to lines 99-102.
Comments 10: L102-103: This is not needed here.
Response 10: We sincerely appreciate your meticulous work and invaluable comments. In accordance with your suggestion, we have deleted this sentence.
Comments 11: L103: Add a hypothesis here.
Response 11: We sincerely appreciate your meticulous work and invaluable comments. In accordance with your suggestion, we have added the classical resource allocation hypothesis. For further details, please refer to lines 106-109.
Comments 12: L107-109: This is in Conclusion.
Response 12: We sincerely appreciate your invaluable comments. In accordance with your suggestion, we have deleted the sentence from the introduction and incorporated it into the conclusion. For further details, please refer to lines 376-375.
Comments 13: L124: Here it is necessary to add information about the population density of the red fox and the Pallas's cat in the study area. Please indicate the known numbers of the two species based on the results of the survey.
Response 13: We sincerely appreciate your meticulous work. We have conducted a comprehensive review of all published literature related to Pallas's cat and red fox within the Sanjiangyuan National Park. Unfortunately, no studies focusing on the population densities of these two small carnivore species were identified.
Comments 14: L126-127: The method of searching for feces should be written in detail. How was the search for feces carried out? Was it a route search or a search by square areas?
Response 14: We sincerely appreciate your meticulous work and invaluable comments. In accordance with your suggestions, we have elaborated on the specific methodologies for sample collection. For further details, please refer to lines 129-135.
Comments 15: L127: Why December and July? This needs an explanation.
Response 15: We sincerely appreciate your valuable feedback. The unique climatic conditions within the Sanjiangyuan National Park divide the year into a cold season (from October to April of the following year) and a warm season (from May to September). During the initial phase of this study's design, our objective was to collect fecal samples of two small carnivore species during both the cold and warm seasons to investigate their dietary differences across seasons. As a result, we collected samples in December 2023 and July 2024, respectively. Unfortunately, due to the limited number of cold-season samples, they lacked sufficient statistical power for meaningful analysis.
Comments 16: L152: Please make the map sector enlargement separately. The map is difficult to interpret in this form.
Response 16: We sincerely appreciate your valuable feedback regarding Figure 1. In accordance with your suggestion, we have revised Figure 1 to improve its academic readability. For further details, please refer to line 160.
Comments 17: L210: The proportion of feces of the two studied species of animals from the total proportion of feces is comparatively small (about 25%). This indicates that most of the feces were collected from other species of animals. In this manuscript, the total volume of feces is not important and, in my opinion, there is no point in indicating it.
Response 17: We sincerely appreciate your meticulous work and invaluable comments. As you have pointed out, the majority of the 160 fecal samples were attributed to other large and small carnivores. In accordance with your suggestion, we have deleted the reference to the total sample size. For further details, please refer to lines 215-219.
Comments 18: L218: The text repeats the data from the figure. They should be presented in a different key so as not to repeat the visual result.
Response 18: We sincerely appreciate your meticulous work. In accordance with your suggestions, we have revised Figure 1 to improve its academic readability. For further details, please refer to lines 225-234. Your suggestions have significantly contributed to enhancing the quality of our research.
Comments 19: L232: The text repeats the data from the figure. They should be presented in a different key so as not to repeat the visual result.
Response 19: We sincerely appreciate your meticulous work. In accordance with your suggestions, we have revised Figure 1 to improve its academic readability. For further details, please refer to lines 249-259.
Comments 20: L245-248: This part requires addition.
Response 20: We sincerely appreciate your meticulous work and invaluable comments. In accordance with your suggestion, we have incorporated the proportions of mammals, birds, and livestock in the diets of the two small carnivores. Additionally, we performed a differential analysis of the feeding proportions of these two small carnivores across different species. For further details, please refer to lines 267-269 and lines 271-275.
Comments 21: L273-275: What do you want to report?
Response 21: We sincerely appreciate your valuable feedback. We have eliminated irrelevant content and restructured this sentence to enhance its contextual coherence and logical flow. For further details, please refer to lines 286-290.
Comments 22: L275-278: This has nothing to do with the manuscript.
Response 22: We sincerely appreciate your insightful suggestion. In accordance with your suggestion, we have eliminated the content that is irrelevant to the manuscript.
Comments 23: L283-286: This should be written about later. Please rearrange the text in a logical sequence of presentation.
Response 23: We sincerely appreciate your meticulous work and invaluable comments. In accordance with your suggestion, we have relocated this sentence to the second paragraph of the Discussion, emphasizing the dietary differences between the two small carnivores. For further details, please refer to lines 291-304. Your suggestions have significantly contributed to enhancing the quality of our research.
Comments 24: L337-340: Is this really necessary to indicate? Since it is not proven, it has no scientific value at this stage of research. And more questions arise.
Response 24: We sincerely appreciate your insightful suggestion. Given that the unidentified species within the Cricetidae family constituted relatively high proportions in the dietary compositions of both Pallas's cats (RRA = 14.39%) and red foxes (RRA = 2.44%), we performed a comprehensive literature review and identified five additional Cricetidae species distributed within the study area. This information has now been integrated into our discussion. In the subsequent stages of our research, we plan to collect additional fecal samples from these two small carnivore species for further validation. Once again, we extend our sincere gratitude for your invaluable suggestions.
Comments 25: L355-363: Provide predator prey stocks and population forecasts to assess predator survival.
Response 25: We sincerely appreciate your meticulous work and invaluable comments. In accordance with your suggestions, we conducted a literature review on the plateau pika within the Sanjiangyuan National Park and discovered that the population density of this species is relatively high. This abundant food resource may alleviate interspecific competition between the Pallas's cat and the red fox to a certain extent. For further details, please refer to lines 387-391. Your suggestions have significantly contributed to enhancing the quality of our research.
Response to Comments on the Quality of English Language
Point 1: The English is fine and does not require any improvement.
Response 1: We sincerely appreciate your recognition of our entire manuscript. We have conducted a comprehensive review of the full text and revised the unclear and erroneous sections to enhance the rigor and readability of the article. All modifications throughout the manuscript are highlighted in red font for clarity.

Round 2
Reviewer 2 Report
Comments and Suggestions for Authors
Seems to be well improved.
Reviewer 3 Report
Comments and Suggestions for Authors
Dear Authors,
The authors tried to make corrections to the manuscript according to the comments. Whether they managed to do this remains debatable. More yes than no. In any case, the article does not bring serious discoveries to science. This is an ordinary comparative study of the diet of two predators. Dietary differences mitigate or do not mitigate interspecies interference - competition between sympatric Pallas's cats and foxes has not been fully proven. But prerequisites for this have been made. Therefore, I leave the decision to the editor. Nevertheless, the manuscript formally meets the main publication criteria. The review of the research results was selected accordingly. The article takes into account the comments on the methodology. The conclusion is supplemented, but without a practical emphasis. Therefore, an average manuscript in terms of significance in science, an article can be published with the comprehensive approval of all members of the review team. In this case, I agree.